# Local Maxima in Martensite Start Temperatures in the Transition Region between Lath and Plate Martensite in Fe-Ni Alloys

**DOI:** 10.3390/ma16041549

**Published:** 2023-02-13

**Authors:** Pascal Thome, Mike Schneider, Victoria A. Yardley, Eric J. Payton, Gunther Eggeler

**Affiliations:** 1Institute for Materials, Ruhr-University Bochum, Universitätsstr. 150, 44801 Bochum, Germany; 2Chimie ParisTech, CNRS, Institut de Recherche de Chimie Paris, PSL University, 75005 Paris, France; 3Department of Mechanical Engineering, South Kensington Campus, Imperial College London, London SW7 2AZ, UK; 4Department of Mechanical and Materials Engineering, University of Cincinnati, Cincinnati, OH 45221, USA

**Keywords:** Fe-Ni alloys, martensitic microstructures, martensite start temperature, lath martensite, plate martensite, transition martensite

## Abstract

In the binary Fe-rich Fe-Ni system, martensite start temperatures M_S_ decrease from 500 to 200 K when Ni concentrations increase from 20 to 30 at.%. It is well known that alloys with Ni concentrations below 28.5 at.% exhibit lath martensite (LM) microstructures (athermal transformation, small crystals, accommodation by dislocations). Above this concentration, plate martensite (PM) forms (burst-like transformation, large crystals, accommodation by twins). The present work is based on a combination of (i) ingot metallurgy for the manufacturing of Fe-Ni alloys with varying Ni-concentrations, (ii) thermal analysis to measure phase transformation temperatures with a special focus on M_S,_ and (iii) analytical orientation imaging scanning electron microscopy for a quantitative description of microstructures and crystallographic features. For Ni-concentrations close to 28.5 at.%, the descending M_S_-curve shows a local maximum, which has been overlooked in prior works. Beyond the local maximum, M_S_ temperatures decrease again and follow the overall trend. The local maximum is associated with the formation of transition martensite (TM) microstructure, which exhibits LM and PM features. TM forms at higher M_S_ temperatures, as it is accommodated by simultaneous twinning and dislocation slip. An adopted version of the Clausius-Clapeyron equation explains the correlation between simultaneous accommodation and increased transformation temperatures.

## 1. Introduction

Since the beginning of systematic materials research, Fe-Ni alloys have received attention for several reasons. Firstly, Fe-Ni alloys with Ni-concentrations close to 36 weight percent, today known as Invar alloys, exhibit a very low thermal expansion coefficient. This phenomenon was discovered by the Swiss physicist Charles Edouard Guillaume in 1895. Invar wire became famous when on 8 June 1912 it was used to measure the thermal expansion of the Eiffel tower which grew and shrank by a few centimeters as the temperature increased and decreased during the day. Guillaume received a Nobel prize in 1920 for his discovery [1]. Secondly, by the beginning of the 20th century, Ni had become an important alloying element in steels, where it stabilizes the γ-phase and improves the high-temperature strength and corrosion resistance [2,3]. Thirdly, the Fe-Ni system is an important reference system in martensite research [4,5,6], where phase transformation temperatures, and especially M_S_ temperatures at which the martensitic transformation starts on cooling from high temperatures, are of central importance. Fe-Ni alloys are known to form two major types of martensite microstructures, lath martensite (LM, below 28.5 at.% Ni) and plate martensite (PM, above 28.5 at.% Ni). Excellent quantitative metallographic descriptions of LM and PM microstructures have been given in the literature (e.g., [7,8]). LM consists of fine longitudinal martensite crystals (laths), with regions of parallel crystallites and a homogenous distribution of lath sizes throughout the microstructure (e.g., [9,10,11,12,13,14,15]). Plate martensite, on the other hand, exhibits a coarser structure with large, sharp-edged lenticular plates and a more heterogeneous size distribution (e.g., [6,10,16,17,18]).

It has been known for a long time that the Fe-Ni system features a concentration-dependent diffusionless transformation [19,20]. Hansen and Anderko documented this tendency for a diffusionless transformation from γ to α’ in their seminal book on phase diagrams [19], reproduced in Figure 1a. Figure 1b shows a plot of the martensite start temperatures M_S_ as a function of Ni-concentrations which was published by Owen in 1990 as a compilation of data from different laboratories [21]. Figure 1b shows an extended version of the Owen plot which includes results from the present work, where M_S_ decreases with increasing Ni-content and at a composition of 28.5 at.% Ni, the curve representing the dependence of the Curie temperature T_C_ on Ni-concentration intersects the M_S_ curve. The present work focuses on the transition region from lath to plate martensite, which has only rarely been studied to date [22].

Research on martensite has a long tradition and the martensitic transformation (MT) features many fascinating thermodynamic, kinetic, micromechanical, chemical/atomistic, and microstructural aspects which have been intensively studied by materials scientists (e.g., [5,29,30,31,32,33,34,35,36,37,38,39,40,41,42,43,44,45,46,47,48,49,50,51,52,53,54]). During the martensitic transformation, a high-temperature phase (austenite, parent phase) transforms into a low-temperature phase (martensite, product phase). On cooling from the high-temperature regime, austenite starts to transform into martensite at the martensite start temperatures M_S_. Martensite formation is completed at the martensite finish temperatures M_F_. It is well known that the MT is a diffusionless, displacive transformation. It proceeds by a lattice shear process, during which each atom retains its nearest neighbors. The alloy Fe-22Ni shows an unusual type of transformation behavior at low cooling rates (0.1 K/min), see [55,56]. Multiple consecutive autocatalytic burst-like transformation steps are observed, which are counteracted and interrupted by increasing interfacial energies and misfit strain energies associated with the formation of martensite laths.

Crystallographic orientation relationships between austenite and martensite have recently been discussed [8,57] and it has been shown how the austenite microstructure can be reconstructed from the martensite data using orientation imaging scanning electron microscopy (OIM, based on electron backscatter diffraction, EBSD). It has been previously shown how this method can be used to obtain high-quality images from martensite microstructures [8]. This method allows differentiation between different types of martensite through analysis of the details of the distributions of three deviation angles (the ξ-angles [8,57]). In the present work, the technique [8,57] is applied to obtain microstructural and crystallographic information from the transition region between lath and plate martensite. The present study was motivated by an unexpected finding in a binary Fe-Ni alloy with 28.5 at.% Ni. For this concentration, we observe a small but significant deviation from the well-known trend that M_S_ temperatures decrease monotonically from approximately 500 to 200 K when the Ni-concentration increases from 20 to 30 at.%. At 28.5 at.% Ni, the curve shows a local maximum that has been overlooked so far. In the present work, ingot metallurgy is combined with thermal analysis and OIM to study this phenomenon. An approach based on a simplified thermodynamic/mechanical model is capable to explain the findings.

## 2. Materials and Methods

### 2.1. Ingot Metallurgy and Scanning Electron Microscopy

The Fe-Ni alloys investigated in the present work were produced using the arc-melting procedure which has previously been used to produce high-purity Ni-Ti alloys [58,59]. Melting was performed using Fe and Ni feedstock in a Buehler AM arc melter (Edmund Buehler GmbH, Bodelshausen, Germany). Fe was obtained from HMW Hauner GmbH&Co. KG (Roettenbach, Germany) in the form of granulate (pieces < 20 mm) with a purity of 99.99 wt.%. HMW also provided the Ni in form of granulate (pieces < 20 mm) with a purity of 99.98 wt.%. The Fe:Ni ratios were controlled using an analytical balance (Mettler Toledo LLC, Columbus, OH, USA) with an accuracy of ±0.001 g. Prior to the first melting step, the reaction chamber was evacuated (vacuum: 3 × 10^−3^ bar) and filled with Ar (purity: 99.9999 vol.%, pressure: 0.6 bar). This cycle was repeated three times. The chamber of the arc melter contained a Ti getter, which was melted for 60 s, to minimize the content of residual oxygen. Subsequently, twelve re-melting cycles and a final drop-casting step were applied. The final ingots had masses of ~50 g, rectangular cross-sections of 8 × 20 mm^2^, and lengths of 60 mm. Casting was followed by hot and cold rolling in a rolling mill of type DWU-30 (Friedrich Krollmann GmbH, Altena, Germany). During six hot rolling steps (with intermediate short-time anneals at 1073 K) the cross section was reduced from 8 to 4.5 mm thickness. Subsequently, during eight cold-rolling steps, the cross-sectional thickness was further reduced from 4.5 to 2.5 mm. The total degree of rolling deformation prior to the subsequent homogenization/austenitization heat treatment was 58.8%. The thermomechanical treated materials were subsequently sealed into evacuated quartz capsules under a vacuum of around 5 × 10^−5^ mbar along with a Ti getter. High-temperature annealings (for homogenization and austenitization) were conducted for 72 h at 1473 K, followed by water quenching, which resulted in an austenitic material state. Small (40 to 50 mg) samples were prepared by cutting, grinding, and polishing (down to a grit size of 1000) and cleaned for 10 min in an ethanol ultrasonic bath. The processing route applied in the present work is documented in Figure 2.

Using this procedure, a total number of eleven ingots was produced. Their target compositions, together with the actual compositions measured after processing using energy dispersive X-ray analysis (EDX) in the SEM and inductively coupled plasma-optical emission spectrometry (ICP-OES), are given in Table 1 for six of the twelve casted ingots. Please note that the experimentally determined concentrations (columns 3 and 4 of Table 1) are very close to the target compositions (column 2 of Table 1). From these eleven ingots, 29 specimens were cut for further investigations. After processing, all ingots were in the austenitic state, with an austenite grain size in the mm range.

### 2.2. Analytical and Orientation Imaging Scanning Electron Microscopy (SEM)

EDX experiments were performed using a LaB_6_ JEOL JSM 840A type SEM (Tokyo, Japan) operating at 30 kV. EDX maps were recorded from areas of 0.25 mm^2^. To obtain high-quality images of martensitic microstructures and to study the distributions of the ξ-angles to differentiate between different types of martensite, an FEI Quanta FEG 650 SEM (Hillsboro, OR, USA) was used. During the EBSD measurements, the specimens were tilted by 70° and an acceleration voltage of 30 kV was used at a working distance of ~17 mm. Investigations were performed on vibro-polished metallographic cross-sections. Areas of 100 × 100 µm^2^ in the center of large prior austenite grains (~mm size) were scanned using a step size of 100 nm resulting in 500,000 crystallographic data points per material state. EBSD measurements were conducted using an EDAX Inc. type Hikari XP camera (Mahwah, NJ, USA). The EBSD data were evaluated following the approach which was recently documented in Ref. [8] and implemented into the MATLAB [60] toolbox MTEX [61,62]. In the present work, two procedures are used which were introduced in previous studies, see Ref. [8]. First, the distribution of the three ξ-angles, which represents the deviation between the actual orientation relationship between the austenite parent lattice and the ideal Bain orientation relation is considered. The three angles are denoted as ξ_1_, ξ_2,_ and ξ_3_, where indices _1_, _2,_ and _3_ correspond to the x, y, and z-axes of the Bain cell respectively., see Figure 3a. The symmetry of the austenite-martensite transformation yields 24 possible martensite variants which can form in one austenite grain. Secondly, a martensite variant color coding which was introduced in [8], Figure 3b, is applied to visualize information about the emerging variants forming during the martensitic transformation.

### 2.3. Thermal Analysis

The martensitic transformation was investigated using differential scanning calorimetry (DSC) in an instrument of type Netzsch DSC 204 F1 Phoenix (Netzsch GmbH, Selb, Germany), following a procedure described elsewhere [58,63]. Small (40 to 50 mg) samples as shown in Figure 2f were prepared by cutting, grinding, and polishing (down to a grit size of 1000) and cleaned for 10 min in an ethanol ultrasonic bath. DSC experiments started at 890 K, after specimens had been thermally equilibrated for 10 min. From 890 K, DSC specimens were cooled down to 88 K at a cooling rate of 10 K/min. For each material state, three individual DSC measurements were performed. In addition, a few individual experiments were performed at cooling rates of 5 and 1 K/min (one test per material state).

## 3. Results

### 3.1. Alloy Chemistry and Martensitic Transformation Temperatures M_S_ and M_F_

In Figure 4 results for two materials, namely Fe-27.5Ni and Fe-30.0Ni are presented, which are associated with LM (Fe-27.5Ni) and PM (Fe-30.0Ni) formation. For the alloy Fe-27.5Ni, DSC specimens were taken from two ingots, Fe-27.5Ni-I, and Fe-27.5Ni-II. Four DSC curves are presented in Figure 4a for this LM alloy, one from Fe-27.5Ni-I and three from Fe-27.5Ni-II. As can be seen in Figure 4a, some scatter in transformation temperatures is present. For the three specimens taken from Fe-27.5Ni-II, this scatter is small. The DSC chart from Fe-27.5Ni-I yields transformation temperatures that are 15 K higher than in Fe-27.5Ni-II. The degree of scatter between different ingots of the same nominal composition was found to be in the order of 10–20 K. In Figure 4a, for the DSC chart of Fe-27.5Ni-I, dashed and dotted reference lines are used to illustrate how M_S_ and M_F_ temperatures were determined (tangent method, see Refs. [62,63]). Likewise, DSC specimens from two ingots were considered for Fe-30.0Ni (Fe-30.0Ni-I, only one of three DSC charts shown, and Fe-30.0Ni-II, all three DSC charts recorded for this material state shown), see Figure 4b. One can see that the formation of PM is associated with a scatter in the range of 40 K (LM scatter range: 15 K). Figure 4c shows that the two types of transformations can be clearly distinguished. Fe-27.5Ni transforms at significantly higher temperatures and features wider transformation intervals and a lower maximum heat flow. Fe-30.0Ni shows much sharper DSC peaks and higher maximum heat flows. However, as can be seen in Figure 4d (data from Fe-27.5Ni-I and Fe-30Ni-I), the latent heats associated with the two types of transformations (areas under blue and red curves) have a similar magnitude. The areas under the two peaks allow latent heats of 1.1 kJ/mol and 1.4 kJ/mol to be evaluated for PM (Fe-30Ni-I) and LM (Fe-27.5Ni-I), respectively.

In Figure 5a,b, the martensite start M_S_ (red symbols and trend lines) and martensite finish M_F_ temperatures (blue symbols and trend lines) are shown as a function of Ni-concentrations for all alloys given in Table 1. Three experiments were conducted per alloy composition. The three individual results for a single composition are often so close that they overlap and cannot be visually distinguished in Figure 5. For the three experiments per material state different symbols (circles, triangles, and crosses) are used. In cases where a second ingot was used, the corresponding data are presented as full diamonds. The dashed vertical line at 28.5 at.% Ni indicates the Ni-concentration which, according to the martensite literature [64,65,66], separates alloys that form LM (lower Ni-concentrations) from those forming PM (higher Ni-concentrations). Below 28.5 at.% Ni, the experimental scatter is negligibly small. There is a higher scatter in the area highlighted by the dashed red rectangle in Figure 5a. This transition region is shown at a higher magnification in Figure 5b. Red and blue trend lines are shown, which are based on the mean values from the three individual DSC experiments.

The data presented in Figure 5a follow a clear overall trend that is consistent with existing literature: *Martensite start temperatures decrease from around 500 to 200 K when the Ni-concentrations increase from 20 to 30 at.%*. Taking a closer look at the transition region, however, a deviation from this trend can be seen, see Figure 5b: Between 28 and 29 at.% Ni, right at the transition between lath and plate martensite, there is a local increase of M_S_ with increasing Ni-concentration. The presence of this local maximum has not been reported so far.

### 3.2. Analysis of DSC Curve Features

To explore the transformation behavior in the transition region, additional DSC experiments were conducted, see Figure 6. The first set of experiments provides insights into the stochastic nature of the martensitic transformation, see Figure 6a. Three DSC charts are shown which were recorded for alloy Fe-28.5Ni at a cooling rate of 10 K/min. Figure 6a documents the stochastic nature of the transformation behavior in the transition zone. Different experiments vary in terms of phase transformation temperatures and show distinctly different DSC chart features. The results for alloy Fe-28.5Ni cannot be rationalized based on one of the two well-defined microstructural transformation scenarios introduced in Figure 4a (LM, smooth and broad DSC curves) and Figure 4b (PM, sharp and narrow DSC peaks).

Specimens Fe-28.5Ni-1 and Fe-28.5Ni-3 start to transform at 266 K. On cooling, specimen Fe-28.5Ni-3 (red DSC chart) shows a gradual increase in heat flux. Then two distinguishable small local maxima are observed, which precede a wide transformation region extending over an interval of almost 50 K. On cooling, specimen Fe-28.5Ni-1 (green in DSC charts in Figure 6a) starts with a gradual increase in heat flux. Then a steep sudden increase typical for PM transformations is observed. This sharp peak is followed by a steady gradual decrease over a wide temperature interval, which converges with the DSC chart of specimen Fe-28.5Ni-3. The third DSC specimen, Fe-28.5Ni-2 (blue curve) shows a clean PM type of transformation behavior, with only one sharp and intense narrow peak. Figure 6b shows a second set of three DSC charts, for alloys with Ni-compositions varying from 28.0 to 29.0 at.% Ni being cooled at a rate of 10 K/min. The specimen with 28 at.% Ni shows a pure lath type of transformation behavior. In contrast, the alloy with 29 at.% Ni features a sharp PM type of DSC peak. Both transformation temperatures follow the overall trend (decreasing transformation temperatures with increasing Ni content). In contrast, specimen Fe-28.5Ni features a DSC chart that cannot be directly associated with one of the two pure types of transformations. It is now interesting to investigate the transformation behavior of alloys with transition-zone compositions at lower cooling rates of 5 and 1 K/min, see Figure 6c,d. At a cooling rate of 5 K/min, the alloys Fe-27.5Ni and Fe-29.5Ni transform showing DSC chart features that can be clearly attributed to LM (Fe-27.5Ni) and PM (Fe-29.5Ni) transformations. In alloy Fe-28.5Ni, the transformation starts at similar temperatures to the Fe-27.5Ni (pure lath type of transformation), but then shows two closely spaced sharp DSC peaks typical for the formation of PM, followed by a short period of gradually decreasing heat fluxes. The DSC charts of Figure 6d were recorded at a cooling rate of 1 K/min. Note that the y-axis has been adjusted to allow the lower heat fluxes associated with Fe-27.5Ni and Fe-28.5Ni transformations to be appreciated. The y-axis is interrupted to capture the maximum of the PM peak observed for Fe-29.5Ni. Figure 4d and Figure 6a schematically illustrate the evaluation of the latent heat from the areas under the DSC charts. Table 2 lists the ΔH_(γ→α’)_ values that were extracted from DSC experiments performed in the present work.

### 3.3. Three Types of Martensite Microstructures

In the following, the application of the orientation imaging SEM procedure outlined in Ref. [8] to study the three types of martensite microstructures is described, see Figure 7. The upper row of Figure 7 (Figure 7a–c) shows color-coded presentations of variant microstructures (as introduced in Figure 3). The lower row (Figure 7d–f) shows the corresponding misorientation angles between the average and local martensite variant orientation (color coding as indicated below the figure). Figure 7a,d show results obtained for a system showing the lath type of transformation behavior (Fe-28.0Ni). The microstructure consists of fine, needle-like martensite crystals, arranged in blocks with highly serrated block boundaries, one of which is highlighted with a white arrow. Figure 7d shows that this LM exhibits a finely dispersed distribution of short-range deviations from the average orientation relationship. The results for the pure PM in Fe-29.0Ni are presented in Figure 7b,e. The PM exhibits some larger crystallites with straight, non-parallel boundaries, one highlighted with a white arrow in Figure 7b, together with smaller crystals occupying the spaces between them. In the PM microstructure of Figure 7e one can see midribs, which are marked with white arrows. This plot also shows that angular deviations increase as one moves from the center of a plate toward its outer boundary. In Figure 7c,f, results for alloy Fe-28.5Ni from the transition region are presented. At first sight, the microstructure shows many features which are characteristic of LM (small martensite crystallites, irregular interfaces between blocks of laths). However, the microstructure also has features in common with PM. For example, boundaries that are long and straight and clearly exhibit plate characteristics. The deviation plot in Figure 7f shows that this transition martensite (TM) microstructure is coarser than in the case of LM and finer than what is observed in PM. Groups of low-deviation regions are observed in the TM structure (blue in the color coding, two blue islands marked by white arrows), forming a long-range microstructural modulation that is not observed in LM.

### 3.4. Statistical Analysis of the OR

Figure 8 shows the relative frequency distributions of the three characteristic angles ξ_1_, ξ_2,_ and ξ_3_, determined following the procedure outlined in Ref. [8]. The distributions of the ξ angles are smooth and have a near Gaussian appearance. However, one can identify differences in the ξ angle distributions between the three martensite microstructures. For all three distributions, the angular position of the maximum and the full width at half maximum (FWHM) are quantified. As far as ξ_1_ is concerned, one finds no significant differences in terms of angular distances from zero. However, the LM peak is smooth and broad, see Figure 8a. In contrast, the PM shows a narrow and sharp peak, see Figure 8b. The transition martensite TM exhibits some features which are similar to those of LM (broad distribution) but also shows a maximum region that is not smooth but exhibits small, sharper maxima, Figure 8c.

For the angles, ξ_2_ and ξ_3,_ one finds that FWHM_LM_ > FWHM_TM_ > FWHM_PM_. When taking the angular maxima positions of LM as a reference, one can see those maxima for the plate martensite are slightly shifted to the right while the angular positions of the maxima of the distributions measured for the TM are only slightly smaller than those observed for LM. Figure 9 shows pole figures, presented in an [100] austenite reference system. Figure 9a–c show the measured [100] α’-directions of all detected MVs (black data points forming data clouds). The figures include the [100] α’-directions corresponding to the three prominent martensite orientation relationships according to Kurdjumov and Sachs (KS, green points [32]) and Nishiyama and Wassermann (NW, red points [33,34]) ORs and Greninger-Troiano OR (GT, blue points [67]).

## 4. Discussion

### 4.1. Three Types of Martensitic Microstructures

So far, the Fe-Ni system has been known to form two types of martensitic microstructures: LM for c_Ni_ < 28.5 at.% and PM when c_Ni_ > 28.5 at.%. When Förster and Scheil discovered these two types of martensite microstructures in the Fe-Ni system, they referred to them as *Umklapp*- (English translation: *folding martensite*, for PM) and *Schermartensit* (English translation: *shear martensite*, for LM) [64,65]. In the English literature, the terms acicular (for PM) and massive (for LM) martensite have also been used [66]. PM and LM can be easily distinguished. Firstly, there is a difference in the thermal transformation behavior during continuous cooling, see Figure 4. LM features higher M_S_ temperatures in a small range (ΔM_S_ = 10 K) and exhibits smooth and broad transformation peaks (full width at half maximum, FWHM = 10 K), Figure 4a. In contrast, PM forms at lower M_S_ temperatures which can show considerable scatter (ΔMs = 40 K), and the associated DSC peaks are sharp (FWHM = 5 K), Figure 4b. Secondly, the microstructural differences between LM and PM are immediately apparent. The LM microstructure consists of blocks with serrated block boundaries. The blocks contain small elongated and parallel crystals, Figure 7a,d. Elementary accommodation processes which accompany the nucleation and growth of LM rely on dislocation plasticity, as can be seen from the high degree of localized misorientations within individual laths. In contrast, PM consists of larger, plate-like martensite crystals, see Figure 7b,e. It is generally accepted that plates can grow very fast (106 mm/s) [68,69]. The first plate forms in the middle of one austenite grain, starting and ending at its grain boundaries, and does not extend across grain boundaries. It divides the grain into two pockets [68,69]. Smaller plates with different variant orientations can form in these pockets. The transformation proceeds by progressive partitioning giving smaller and smaller pockets [68,69]. Plates can also form in an autocatalytic process, forming zig-zag arrays [70]. They can be twinned, untwinned, and partially twinned. In the latter case, one can observe midribs [70], see Figure 7e. It is generally accepted that twinning is the dominant accommodation process when PM forms. SEM-OIM micrographs for PM are shown in Figure 7b and e. LM and PM not only differ in thermal transformation behavior and microstructural features but also show small but significant differences in the distributions of ξ angles (LM: broad distributions, PM: sharp distributions), see Figure 8. An excellent description of LM and PM microstructures in Fe-based alloys have been given by Maki [7]. In the present work, a martensitic transformation was observed which exhibits thermal, microstructural, and crystallographic features with contributions from both LM and PM. We refer to this microstructure as transition martensite (TM). Only Wang and Ma [22] have previously reported a “mixed martensite morphology” in the Fe-Ni system in 1987, which they did not relate to M_S_ temperatures. Figure 7c,f show that the microstructure of TM exhibits PM and LM features. Due to the inherent heterogeneity and complexity of martensitic microstructures, one can easily overlook the specific features of this microstructure, which is observed for Fe-Ni alloys with 28.5 at.% Ni. This TM is important because it forms in the compositional region where the small local maximum in the overall descending M_S_(c_Ni_)-curve is observed.

### 4.2. Transformation Heats

Kaufmann and Cohen [4,71] were the first to take a closer look at heat effects and driving forces associated with martensitic transformations in the Fe-Ni system. One of their conclusions was [71] that a driving force above 1.3 kJ/mol is required to trigger the formation of PM. Driving forces increase with increasing Ni-concentrations and are high enough to form plate martensite when the alloy composition reaches 28 at.% Ni [72]. In their review from 1992, Kaufmann and Hillert [73] pointed out that calculated and experimental values for heats of transformation for an alloy composition of 25 at.% Ni shows reasonable agreement (calculated value: 3.321 kJ/mol [4,71], experimental value: 3.362 kJ/mol [15,74]). In their textbook on phase transformations, Porter et al. [47] report 1.930 kJ/mol for a Fe-Ni alloy with a Ni-content of 28 at.% Ni. The scatter in the data reported in the literature can be as high as a factor of 3 [71,73]. The values listed in Table 2 are only a little below the scatter band reported in the literature. While the present study was not designed to measure precise values of latent heats, the results shown in Table 2 allow us to conclude that the local increase of M_S_ in the transition region is not related to the local change in latent heats. This is also in line with relatively small changes in values reported by Kaufmann and Cohen [71] when the Ni-content increases from 27.5 to 29.5 at.%.

### 4.3. Kinetic Nature of Martensitic Transformations

Three types of transformation mechanisms have been proposed to account for observed differences in martensitic transformations: (i) the athermal, (ii) the isothermal, and (iii) the burst mode [69]. The LM which forms in the Fe-Ni system for Ni-compositions of 27.5 at.% can be attributed to the athermal mode. Even when specimens are taken from different ingots, the scatter is small, see Figure 4. In the athermal mode, the amount of martensite formed is a function only of the temperature; holding times have no effect and there is no significant influence from the cooling rate, Figure 6b–d. A comparison of the results shown in Figure 6c,d does not allow us to fully exclude an effect of cooling rate, but it can be clearly seen that it is at best weak. While the effect is not significant in view of the experimentally observed scatter, the fact that the peaks are narrower is noteworthy. One also may conclude that the beginning of the transformation associated with the formation of LM for specimen Fe-27.5Ni is slightly higher for the lower cooling rate and that there are finer serrations in the DSC chart of the Fe-28.5Ni specimen. However, these effects are also small and thus it can be concluded that the cooling rates used in the DSC experiments do not significantly affect the features of the transformations.

While it is not uncommon to find retained austenite below M_F_, no residual austenite was detected in the material states studied in the present work. The sharp and intense DSC peaks in Figure 4b represent experimental evidence for the burst-like martensitic transformation from austenite to PM [47]. In this burst mode, a large number of plates form in a short time interval, in a chain reaction of autocatalytic nature. Figure 4b,c show that the formation of PM is associated with sharp DSC peaks in a 40 K temperature interval. The MT has a strong first-order character and the minimization of the elastic strain energy during the transformation is of utmost importance [9,39,71]. This is why during both nucleation and growth of martensite, interaction with defects needs to be considered [44,72,75,76,77,78,79,80]. The nucleation of lath martensite, which shows little scatter in M_S_ temperatures, is associated with accommodation by dislocations and it is well-known that even well-annealed crystals contain dislocation densities in the order of 1014 m^−2^, which results in free paths between individual dislocations which are smaller than 1 µm. It seems reasonable to assume that when such readily available defects act as nucleation sites, M_S_ temperatures show a small scatter. In contrast, plate martensite, accommodated by twinning, nucleates at grain boundaries. Due to the large grain size (~mm range), the DSC specimens only contain a limited number of grain boundaries and hence a much lower density of available nucleation sites. Ueda et al. [81] showed that the character of grain boundaries affects the nucleation of martensite, and thus one can rationalize the larger scatter in M_S_ temperatures when PM forms by nucleation at grain boundaries. In the present work, the possibility that this scatter is related to chemical fluctuations can be excluded, as local compositions were measured using EDX, and no significant differences in chemical compositions were found. The relatively large scatter in the M_S_ temperatures of plate martensite in the Fe-Ni system is yet another example of unavoidable inherent microstructural scatter in important material properties depending on heterogeneous nucleation, which has also been reported for other materials research fields such as in creep damage nucleation in uniaxial creep testing [82] or for the onset of plastic deformation in micro shear tests [83].

### 4.4. A Thermomechanical Explanation for the Local M_S_ Maximum

Figure 10 shows the M_S_ temperatures obtained in the present work presented in Figure 4 together with the trend lines from the extended Owen plot from Figure 1b (dashed grey line: M_S_ data, dotted grey line: Curie temperatures) in the background. It can be seen that the small local M_S_ maximum at a Ni-concentration of 28.5 at.% deviates from the general trend of decreasing M_S_ with increasing Ni-content.

We suggest that the local M_S_ maximum is related to the effect of accommodation processes on the critical lattice shear stress which needs to be overcome to activate the shear process through which austenite transforms into martensite. Accommodation processes help to lower this critical shear stress. It is well accepted that at higher temperatures, dislocation plasticity promotes the nucleation and growth of LM, while at lower temperatures, internal twinning accommodates strain energies associated with the formation of PM [15,22,38,39,47,66,74,84,85,86,87,88]. It has also been suggested that dislocation slip is easier than twinning at higher temperatures [22,66,84]. In Figure 11a the microstructural and thermal results are summarized and the local maximum is highlighted in a plot in which the Ni-concentration is plotted on the y-axis against the M_S_ temperatures on the x-axis, for consistency with the schematic plots of Figure 11b,c (see below).

Hornbogen [39,89] has suggested an equation relating the M_S_ temperatures to mechanical and thermodynamic parameters, which can be derived from the Clausius-Clapeyron equation (e.g., [90]):(1)$$M_{S}\;=\;T_{0}\cdot \left(1-\;\frac{\tau_{\gamma -\alpha \prime }\cdot \phi_{\gamma -\alpha \prime }}{\Delta H_{\gamma -\alpha \prime }}\right)\;$$

In Equation (1), *T_0_* is the temperature at which the phases γ and α’ are in thermodynamic equilibrium, *τ_γ−α’_* is the shear stress which is required to overcome activation barriers associated with the nucleation and growth of martensite, *Φ_γ−α’_* represents the transformation strain and Δ*H_γ−α’_* is the transformation enthalpy. *T*_0_, which cannot be directly measured, governs the overall M_S_ trend shown in Figure 1, but cannot explain the local maximum. The results compiled in Table 2 suggest that the local maximum is not related to Δ*H _γ−α’_*. In the present work, *Φ _γ−α’_* can be considered to be constant. Therefore, it seems reasonable to assume that *τ_γ−α’_*, which is directly related to the resistance of the Fe-Ni system to twinning and dislocation slip, is responsible for the local maximum.

In a first-order approximation, a spring dashpot model shown in Figure 11b can be used to explain the local M_S_ maximum. In this approximation, a system is considered which is just starting to transform. The resulting increase of elastic strain energy is represented by a spring under compressive stress. This stress can be reduced when the two dashpots, representing dislocation plasticity and twinning, start to deform. In Figure 11b, thinner and thicker dashpots represent cases where deformation is easier and more difficult, respectively. As is schematically illustrated in Figure 11c, at low temperatures twinning represents the dominant accommodation process, while dislocation plasticity does not play a role. At high temperatures, the opposite holds true and dislocation plasticity governs stress relaxation. Figure 11c shows that accommodation is easiest and most effective when both processes, i.e., twinning and dislocation slip, yield significant contributions. This is where *τ_γ−α’_* is lowest and where, according to Equation (1), a local maximum in M_S_ temperatures is observed.

### 4.5. Martensitic Transformations, Magnetism, and Atomistic Aspects

Owen [21] pointed out that martensite which formed from ferromagnetic austenite showed a lath structure, while plate martensite formed on cooling from paramagnetic austenite. Owen also suggested [21] that lattice softening of the austenite played a role in the early stages of martensite nucleation, a view that is shared by other martensite researchers [84]. The importance of the magnetic transition in the Fe-Ni system was highlighted by Acet et al. [27] and Xiong et al. [28]. Acet et al. [27] pointed out that in a narrow composition range around c_Ni_ = 30 at.%, the martensitic transformation occurred below the Curie temperature, where the parent phase was magnetically ordered. As can be seen in Figure 10, transition martensite and the associated thermal anomalies are found in a range where the M_S_(c_Ni_) and T_C_(c_Ni_) curves intersect. It is presently unclear whether the magnetic transition in the Fe-Ni system plays a role in the evolution of martensite microstructures, and it remains to be explored whether the magnetic transition is directly related to the microstructural LM-TM-PM transitions. A further consideration is the possible role of an atomic percolation threshold, above which nearest-neighbor interaction can occur [91]. Percolation models consider regular lattices with randomly occupied sites without preferences and with statistically independent probabilities. A critical threshold value of the concentration of the minor component needs to be reached, to enable long-range connectivity between atoms and the formation of clusters that are homogeneous on the sub-µm-scale. In binary systems, many different threshold concentration values for the minor component have been reported, two literature examples are 20 at.% for face-centered cubic [92,93] and 24 at.% for body-centered cubic [94,95] materials. In the present work, the new observation of transition martensite and M_S_ anomaly is made for a concentration of 28.5 at.% Ni, which is somewhat higher than previously reported percolation thresholds. This suggests that a higher percolation threshold concentration is required for the transition from lath to plate martensite to occur.

## 5. Summary and Conclusions

In the present work, the martensitic transformations in the binary Fe-Ni system are systematically characterized. At an intermediate Ni-content, there is a small but significant deviation from the well-known trend of decreasing M_S_ temperatures with increasing Ni-content. From the results obtained in the present work the following conclusions can be drawn:

(1) There is a small but significant deviation from the trend of decreasing M_S_ temperatures (from 500 to 200 K) with increasing Ni-concentrations (from 20 to 30 at.%), which has not been reported so far: Between 28 and 29 at.% Ni a small local maximum in M_S_ temperature is observable.

(2) This transition region separates alloy compositions exhibiting two distinct types of martensite: (i) lath martensite (LM, lower Ni-concentrations, <28.5 at.% Ni, accommodation by dislocation plasticity) and (ii) plate martensite (PM, higher Ni-concentrations, >28.5 at.% Ni, accommodation by twinning) which nucleate and grow on cooling from the austenite regime. The fine LM microstructure consists of small, distorted, and deformed elongated crystals while the coarser PM microstructure consists of large, twinned plates.

(3) The local maximum of the M_S_ (c_Ni_)-curve at 28.5 at.% Ni is associated with a martensitic microstructure which we have denoted as (iii) transition martensite (TM). The TM microstructure shows mixed features from both LM and PM. It is suggested that this type of mixed martensite forms when both types of accommodation processes, i.e., dislocation plasticity and twinning, can operate simultaneously. Dislocation plasticity and twinning are promoted by higher and lower temperatures, respectively. At 28.5 at.% Ni, the transformation occurs at temperatures where both accommodation processes yield significant contributions. This lowers the critical stress required to trigger the martensitic shear process in the lattice and thus results in a small increase in M_S_. The results obtained in the present work suggest that the concentration range over which this can occur is small.

(4) Thermal analysis by means of differential scanning calorimetry shows that lath martensite exhibits smooth and broad peaks, indicating an athermal type of transformation behavior. In contrast, plate martensite shows a burst-like type of transformation, which manifests itself in sharp and narrow DSC peaks. The mixed martensite which forms in the transition region shows DSC charts with both broad and smooth and sharp burst-like features. The transformation enthalpies associated with the three types of martensitic microstructure are in the same order of magnitude and therefore cannot explain why TM nucleates at higher temperatures.

(5) Further work is required to clarify two points: Firstly, whether magnetic effects also play a role in the transformation characteristics, or whether it is a coincidence that the curve describing the concentration dependence of the Curie temperature intersects the M_S_ (c_Ni_)-curve precisely in the composition range where TM forms. Secondly, it should be considered whether the percolation of atoms plays a role.

(6) The three types of martensitic microstructures investigated in the present work can not only be differentiated based on SEM micrographs and DSC chart features, but also by evaluating the Kurdjumov/Sachs ξ-angles using the EBSD data [8]. The distribution of three angles ξ_1_, ξ_2_ and ξ_3_ (where indices _1_, _2,_ and _3_ correspond to the x, y, and z-axes of the Bain cell) shows microstructure-specific distributions. The present crystallographic analysis also shows that a statistically sufficient number of data points scatter around the maxima of orientation distribution functions, such that it is not possible to differentiate between precise crystallographic orientation relationships, which have been proposed in the literature.

(7) It is unclear if the formation of TM influences other properties (e.g., mechanical properties, such as hardness or yield strength) of the investigated alloys. Further work is required to characterize the mechanical and functional properties of the newly observed type of martensite.

## Figures and Tables

**Figure 1 materials-16-01549-f001:**
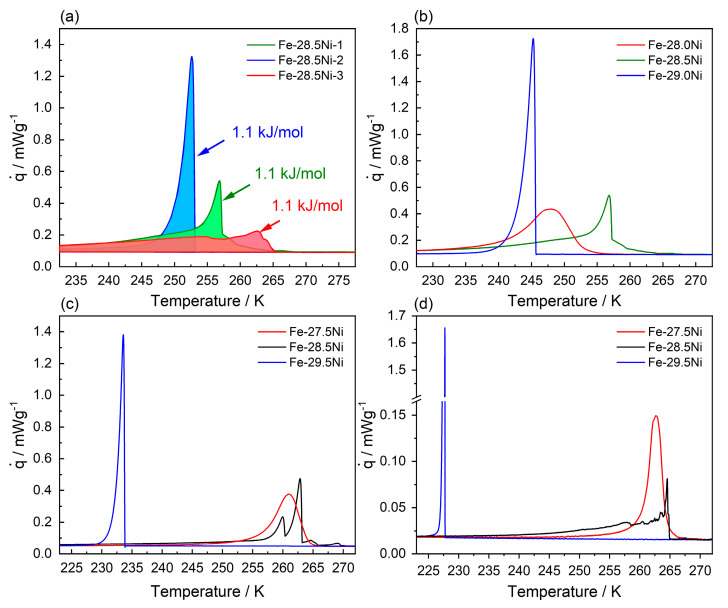
Phase transformations in the Fe-Ni system. (**a**) Phase diagram as reported by Hansen and Anderko [19] extended by details regarding the transition from α to α’ from Swartzendruber et al. [20]. (**b**) Extended Owen plot [21] showing the dependencies of M_S_ and T_C_ on the Ni-concentration in the Fe-Ni system. With data from [22,23,24,25,26,27,28] and results from the present study.

**Figure 2 materials-16-01549-f002:**
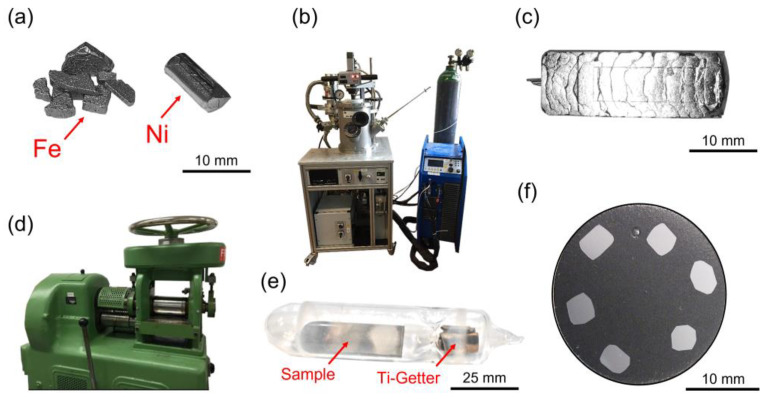
Fe-Ni specimen processing route. (**a**) Raw materials (feedstock). (**b**) Arc melter. (**c**) Drop cast ingot prior to hot and cold rolling. (**d**) Rolling mill. (**e**) Encapsulated specimens prior to heat treatment. (**f**) Embedded DSC/SEM specimens.

**Figure 3 materials-16-01549-f003:**
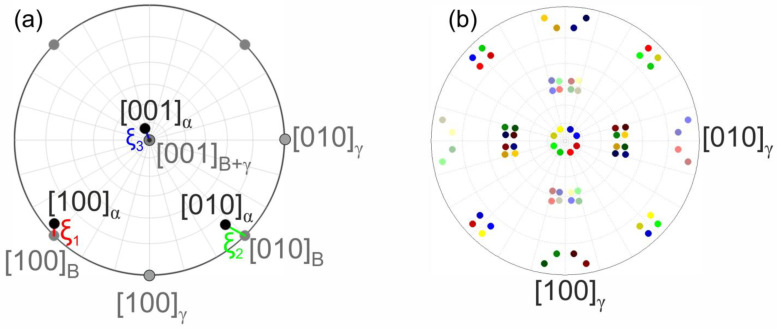
Illustration of two procedures from Ref. [8] used in the present work. (**a**) Illustration of ξ-angles. (**b**) Color coding for 24 martensite variants. B refers to the Bain reference cell and γ and α are the crystal axes of austenite and martensite, respectively.

**Figure 4 materials-16-01549-f004:**
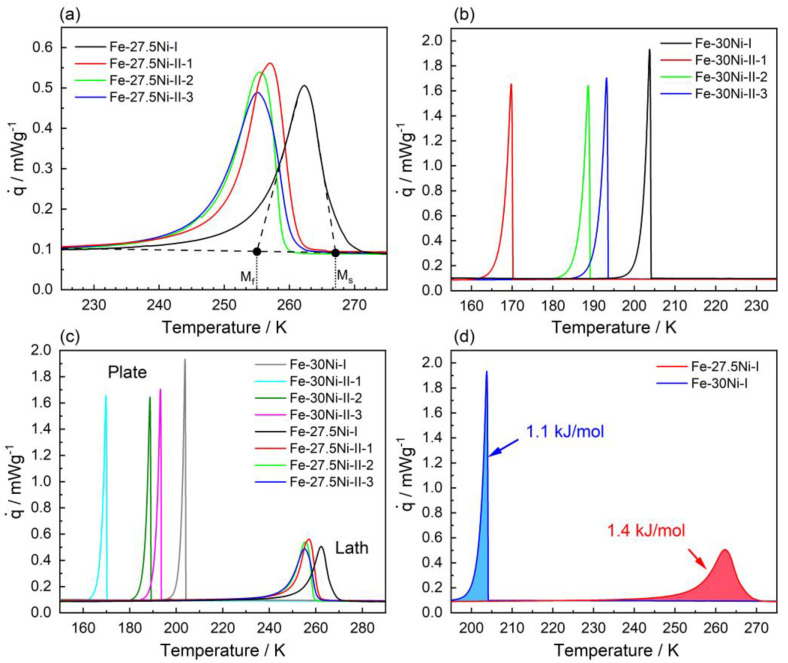
Thermal analysis of the martensitic transformation in Fe-27.5Ni and Fe-30Ni. DSC charts were recorded at a cooling rate of 10 K/min. (**a**) DSC charts recorded for Fe-27.5Ni-I and -II. The tangent method is illustrated for one of the Fe-27.5Ni charts. (**b**) DSC charts recorded for Fe-30Ni-I and -II. (**c**) Plot collating the data from both binary alloys. (**d**) Direct comparison of the latent heats associated with the transformation for Fe-27.5Ni-I and Fe-30Ni-I.

**Figure 5 materials-16-01549-f005:**
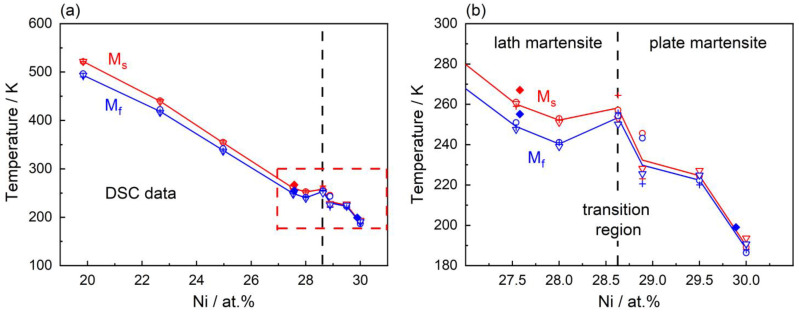
M_S_ and M_F_ temperatures as a function of Ni-concentration. M_S_—red crosses and red trend lines. M_F_—blue crosses and blue trend lines. (**a**) Overall trend. M_S_ temperatures decrease between 20 and 30 at.% Ni. (**b**) Inset of the transition region between 27.5 and 30.0 at.% Ni is highlighted in Figure 5a.

**Figure 6 materials-16-01549-f006:**
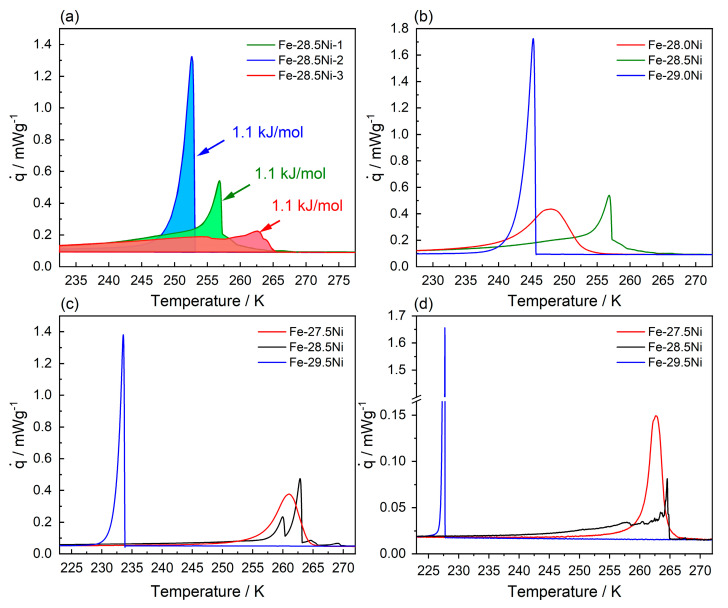
Four sets of DSC charts from the transition region. (**a**) Three specimens of composition Fe-28.5Ni. Latent heats for three experiments (areas under DSC charts) were determined as 1.1 kJ/mol. Specimens were recorded at cooling rates of (**b**) 10 K/min, (**c**) 5 K/min, and (**d**) 1 K/min.

**Figure 7 materials-16-01549-f007:**
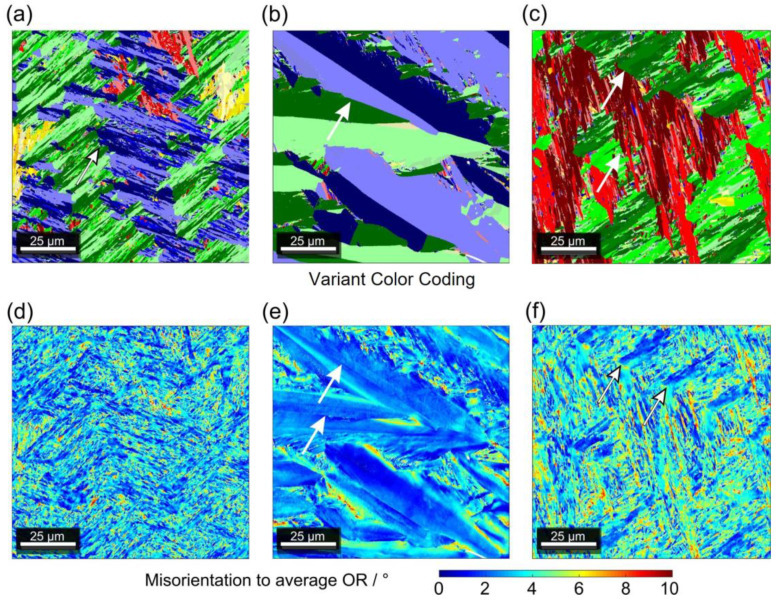
OIM-EBSD results. (**a**–**c**) Color-coded presentations of MVs (see Figure 3b). (**d**–**f**) Color-coded presentation of misorientation angles between measured and average MV orientations. (**a**,**d**) LM observed in Fe-28Ni. (**b**,**e**) PM observed in Fe-29Ni). (**c**,**f**) TM in Fe-28.5Ni.

**Figure 8 materials-16-01549-f008:**
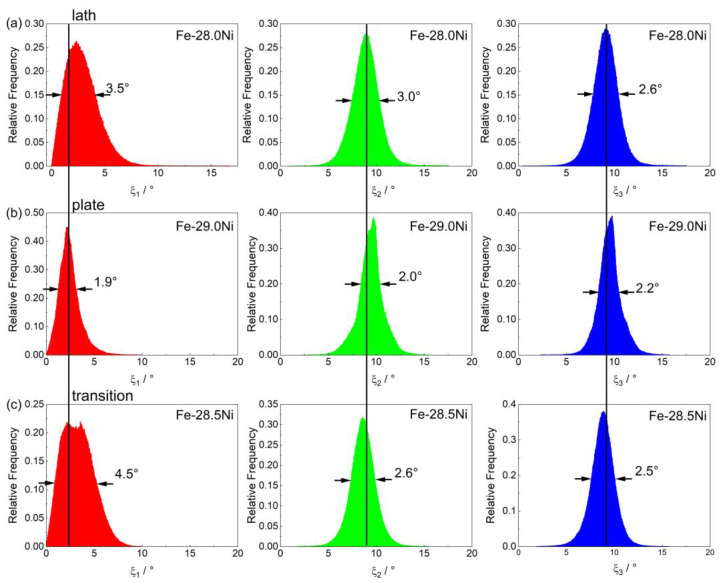
Histograms showing the relative frequency distributions of the characteristic angles ξ_1_ (left column), ξ_2_ (middle column), and ξ_3_ (right column) obtained from the OIM-EBSD data presented in Figure 7. (**a**) Lath martensite (Fe-28.0Ni). (**b**) Plate martensite (Fe-29Ni). (**c**) Martensite forms in the transition region (Fe-28.5Ni).

**Figure 9 materials-16-01549-f009:**
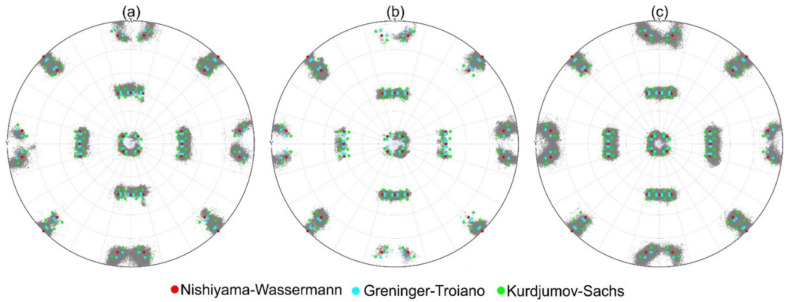
Martensite variant orientations are presented as <100> α’ pole figures in a <100> γ austenite standard projection. In addition to the experimental grey data clouds the ORs of NW (red), GT (light blue), and KS (light green) are presented. (**a**) Lath martensite. (**b**) Plate martensite. (**c**) Transition martensite.

**Figure 10 materials-16-01549-f010:**
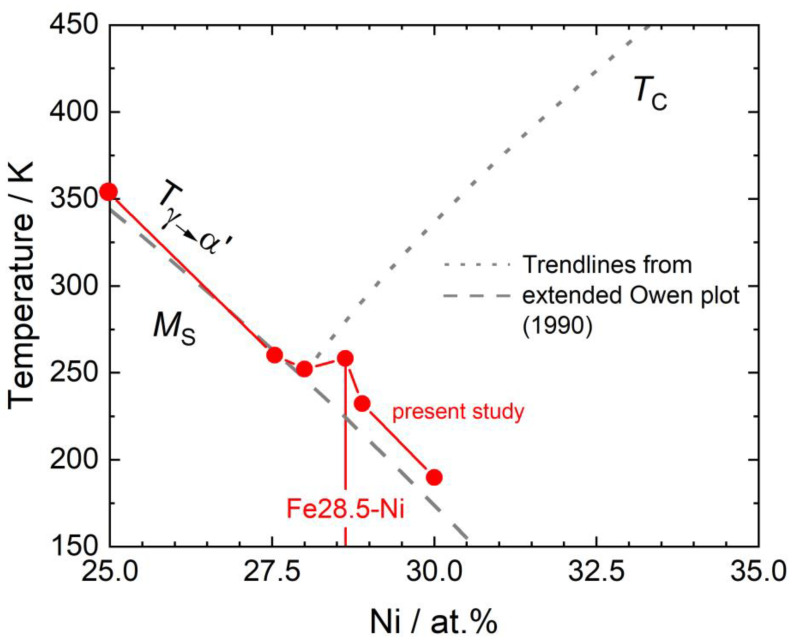
M_S_ temperatures obtained in the present work together with trend lines from the extended Owen plot [21] from Figure 1b.

**Figure 11 materials-16-01549-f011:**
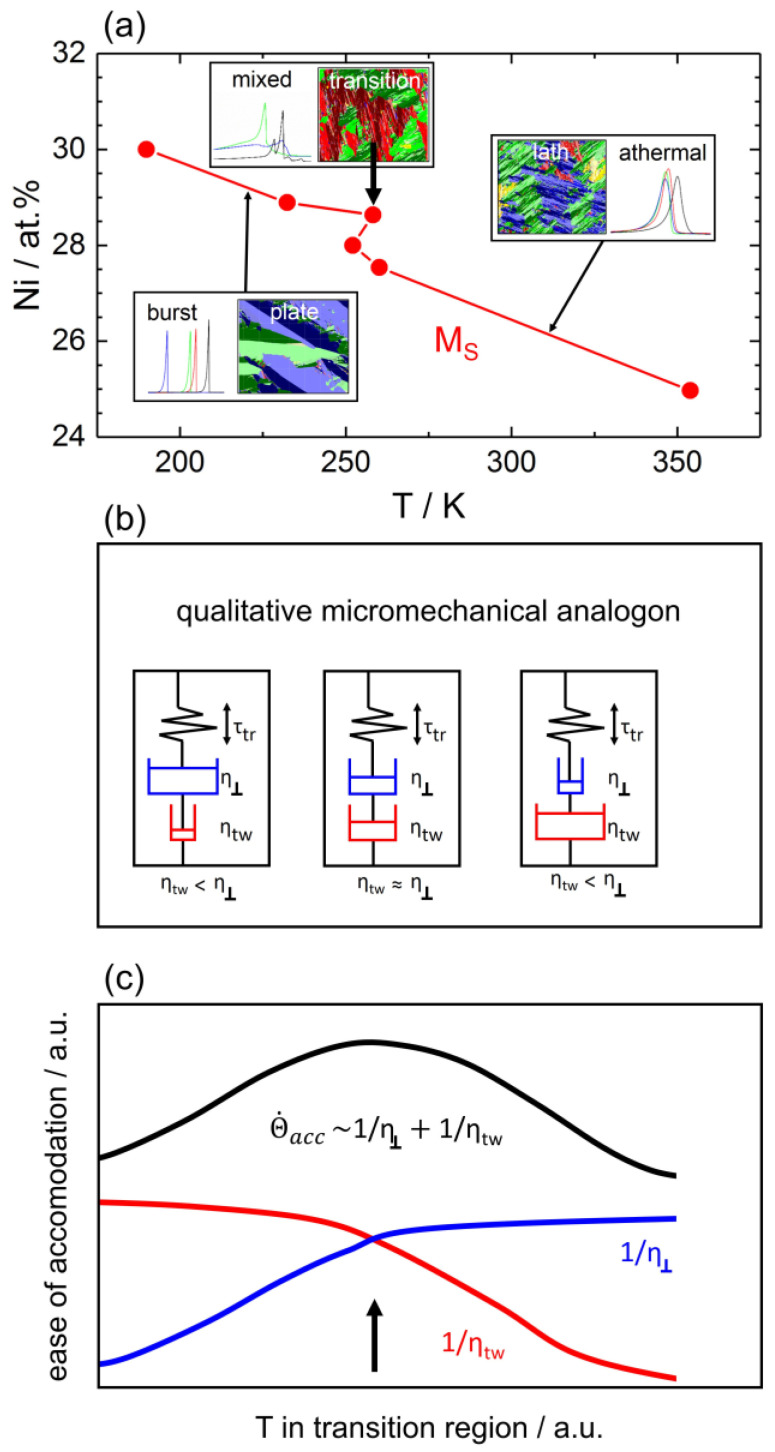
Schematic drawings rationalizing the local MS maximum in the transition region. (**a**) Local maximum in PM/LM transition region highlighted with a thick black arrow pointing down. (**b**) Qualitative micromechanical model. Accommodation can be provided by dislocation plasticity (blue dashpot) and twinning (red dashpot). (**c**) Maximum ease of accommodation is observed when both accommodation processes can operate simultaneously.

**Table 1 materials-16-01549-t001:** Target and experimentally determined compositions (Ni-content in at.%) of the binary alloys. Roman numerals I and II indicate that specimens from two different ingots were considered.

Alloy #	Target Ni Content [at.%]	Ni Content (EDX) [at.%]	Ni Content (ICP-OES) [at.%]
Fe-20.0Ni	20.0	19.8	-
Fe-22.5Ni	22.5	22.7	-
Fe-25.0Ni	25.0	25.0	-
Fe-27.5Ni-I	27.5	27.6	-
Fe-27.5Ni-II	27.5	27.4	27.54
Fe-28.0Ni	28.0	28.0	28.00
Fe-28.5Ni	28.5	28.2	28.63
Fe-29.0Ni	29.0	28.8	28.89
Fe-29.5Ni	29.5	29.3	29.50
Fe-30.0Ni-I	30.0	29.9	-
Fe-30.0Ni-II	30.0	29.7	30.00

**Table 2 materials-16-01549-t002:** Transformation enthalpies in kJ/mol evaluated for alloy compositions between 27.5 and 30.0 at.% Ni from DSC charts recorded at different cooling rates. Three experiments were evaluated at a cooling rate of 10 K/min.

Alloy #	$\Delta H_{\gamma \to \alpha \prime }$ in kJ/mol DSC: 1 K/min	$\Delta H_{\gamma \to \alpha \prime }$ in kJ/mol DSC: 5 K/min	$\Delta H_{\gamma \to \alpha \prime }$ in kJ/mol DSC: 10 K/min
Fe-27.5Ni-II	1.3	1.4	1.4, 1.4, 1.3
Fe-28.0Ni	1.0	1.3	1.2, 1.1, 1.1
Fe-28.5Ni	0.9	1.2	1.1, 1.1, 1.1
Fe-29.0Ni	0.8	1.1	1.1, 1.3, 1.3
Fe-29.5Ni	1.2	1.0	1.2, 1.1, 1.1
Fe-30.0Ni-II	1.1	1.1	1.1, 1.1, 1.1

## Data Availability

Data will be made available on request.

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
