# Peer review of "Local Maxima in Martensite Start Temperatures in the Transition Region between Lath and Plate Martensite in Fe-Ni Alloys"

_materials, 2023, doi:10.3390/ma16041549_

Round 1

Reviewer 1 Report

This paper investigates the influence of alloying element Ni content on martensite start temperatures and its effect on martensitic characteristic in the binary Fe-Ni alloys. The study is interesting and well scripted. and it should be published with minor revision. Kindly consider the following points.

1. A space should appear between the value and unit;

2. The main conclusions or the results should be showed instead of excessive background and experimental method in the abstract. 

3. The singular and plural forms should be unified in the manuscript, such as temperature and temperatures.

4. The most of references are too old and timeworn, up-to-date reference should be adopted. In addition, the formats of the references should exactly match the Journal of Materials.

5. The ultimate goal of adjusting microstucture is controlling properties of the material, in this work, the corresponding properties are never studied, I suggest the authors can add some properties measurements.

Reviewer 2 Report

Dear Editor,

thank You for an opportunuty to review manuscript entitled: “Local Maxima in Martensite Start Temperatures in the Transition Region between Lath and Plate Martensite in Fe-Ni Alloys”, submitted by Pascal Thome, Mike Schneider , Victoria A. Yardley , Eric J. Payton , Gunther Eggele.

Authors have focused on differences in type of martensite in FeNi alloys. Experiment was adequately designed and provides large number of results. Some parts of manuscript (mainly chapter Results) are not described clearly and should be improved). Therefore, I have following suggestions:

GENERAL

          The style of this manuscript must be modified. Authors use terms as “we have…”, “our method …. ”, “we showed …..”, etc. This is not the way how to write scientific paper. Whole text should be “depersonalized”.

          Figure captions contains data already given in figures. Also, in some cases, authors even discuss numerical values from diagrams in captions! These captions should be shortened.

          Description of method used in chapter 2 (lines 181-208 – including Figure 4) is presented via results of this paper!!?!? This part should be moved to chapter 3, Results.

          Key words: dependence on alloy chemistry is not correct. Please modify it.

1.     line 28 – stated “             different.

2.     Line 60 – TC        C to subscript.

3.     Figure 1 contains data (points) from present work?!?!!? You must remove these points from figure 1 and modify the caption.

4.     Figure 2 – instead of “Fe-Ni ingot metallurgy processing route “ should be  Fe-Ni specimen processing route .

5.     Line 139 – eXperimental          x is missing

6.      Figure 3 – add to caption meaning of g and B

7.     Figure 5 – Caption must be rewritten. NO comments on obtained values!

8.     Lines 283-290 – move to chapter Discussion

9.     Line 297 – Table number is 2. Also, delete second sentence of caption

My opinion is that these corrections will stongly improve this manuscript. Finally, I would like to review this paper again, after authors make corrections.
